# Polycystin-2 Mediated Calcium Signalling in the *Dictyostelium* Model for Autosomal Dominant Polycystic Kidney Disease

**DOI:** 10.3390/cells13070610

**Published:** 2024-03-31

**Authors:** Claire Y. Allan, Oana Sanislav, Paul R. Fisher

**Affiliations:** Department of Microbiology, Anatomy, Physiology and Pharmacology, La Trobe University, Bundoora, Melbourne, VIC 3086, Australia; claire.allan@latrobe.edu.au (C.Y.A.);

**Keywords:** ADPKD, Polycystin-2, *Dictyostelium*, calcium, calcium signalling, autophagy

## Abstract

Autosomal dominant polycystic kidney disease (ADPKD) occurs when the proteins Polycystin-1 (PC1, *PKD1*) and Polycystin-2 (PC2, *PKD2*) contain mutations. PC1 is a large membrane receptor that can interact and form a complex with the calcium-permeable cation channel PC2. This complex localizes to the plasma membrane, primary cilia and ER. Dysregulated calcium signalling and consequential alterations in downstream signalling pathways in ADPKD are linked to cyst formation and expansion; however, it is not completely understood how PC1 and PC2 regulate calcium signalling. We have studied Polycystin-2 mediated calcium signalling in the model organism *Dictyostelium discoideum* by overexpressing and knocking down the expression of the endogenous Polycystin-2 homologue, Polycystin-2. Chemoattractant-stimulated cytosolic calcium response magnitudes increased and decreased in overexpression and knockdown strains, respectively, and analysis of the response kinetics indicates that Polycystin-2 is a significant contributor to the control of Ca^2+^ responses. Furthermore, basal cytosolic calcium levels were reduced in Polycystin-2 knockdown transformants. These alterations in Ca^2+^ signalling also impacted other downstream Ca^2+^-sensitive processes including growth rates, endocytosis, stalk cell differentiation and spore viability, indicating that *Dictyostelium* is a useful model to study Polycystin-2 mediated calcium signalling.

## 1. Introduction

Autosomal dominant polycystic kidney disease (ADPKD) is a very frequently occurring renal disease in which people suffer from enlarged renal and hepatic cysts and can often be fatal. Mutations in the genes *PKD1* and *PKD2* encoding the two key proteins, polycystin-1 (PC1) and Polycystin-2 (PC2) respectively, are associated with the disease [1], with most cases (over 85%) linked to PC1 mutations; however, mutations in PC2 also contribute to disease pathology [1,2]. PC2 belongs to the transient receptor potential (TRP) channel superfamily of membrane-embedded ion-conducting channels. TRPs are involved in many cellular processes and are increasingly being investigated for their potential as therapeutic drug targets [3]. In mammals, PC2 is localized to various cellular structures, which include the primary cilium, plasma membrane and endoplasmic reticulum (ER) [4,5,6,7,8,9]. Its role as a mediator of intracellular and extracellular calcium release positions PC2 as a key player in sensory signalling [4,9]. PC2 forms complexes with PC1, an integral membrane mechanosensitive receptor situated in the primary cilium [10]. When subjected to fluid stress, PC1 triggers calcium influx into the cytoplasm through PC2 [6]. This calcium signal is further augmented by the release of calcium from the ER via PC2, thereby governing calcium-dependent signalling processes [6,11,12]. Moreover, PC2 interacts with other intracellular calcium channels, such as the Inositol trisphosphate receptor (InsP3R) of the ER and the ryanodine receptor of the sarcoplasmic reticulum, to regulate calcium signalling [13,14,15]. ADPKD cells exhibit reduced intracellular calcium concentrations, which are believed to be associated with elevated levels of cyclic adenosine monophosphate (cAMP). This dysregulation leads to fluid secretion, activation of MAPK/ERK signalling pathways and subsequent cell proliferation [16]. Ultimately, these disrupted processes result in uncontrolled cell proliferation and the development of fluid-filled cysts [17,18,19]. Cyst formation and progression involve oxidative stress [20,21], inflammation and cell death [22,23]; however, the mechanisms behind cyst development are still poorly understood. 

Extensive research has been conducted on PC2 in various cellular models [24], including *Caenorhabditis elegans* [25], *Drosophila* [26,27] and mice [28]. Additionally, a PC2 homologue called Polycystin-2 [29], also known as PKD2 [30], TrpP [31] or Trpp2 [32], has been identified in the model organism *Dictyostelium discoideum* [29]. The *Dictyostelium* Polycystin-2 protein (gene name *pkd2* or *TrpP*) has been the subject of a small number of studies that have only just begun to understand its function. The utilization of *Dictyostelium* as a model organism for studying ADPKD has been demonstrated in a compelling study in which the antiproliferative drug Naringenin was screened using *Dictyostelium* mutants to assess its impact on cell growth. Remarkably, Naringenin exhibited growth restriction specifically dependent on Polycystin-2, and the study further demonstrated that this growth inhibition also reduced the progression of cyst development in Madin–Darby canine kidney (MDCK) tubule cells in a collagen matrix [33]. These findings emphasize the potential of *Dictyostelium* in investigating ADPKD and highlight the critical need to deepen our understanding of protein function in this organism, especially in relation to its calcium regulatory properties.

*Dictyostelium* is a simple and tractable model that is ideal for studying protein function and the molecular mechanisms underpinning disease pathogenesis. It presents an excellent opportunity to explore the molecular functions of Polycystin-2 further. *Dictyostelium* Polycystin-2, which shares 46% protein similarity with human PC2, is predicted to have six transmembrane domains, a conserved pore region between TM 5 and 6 and a conserved C-terminal coiled-coil domain [30]. Unlike mammalian PC2, Polycystin-2 lacks the EF-hand motif and ER retention signal [29]. Immunolocalization studies with FLAG-tagged Polycystin-2 show its presence mainly at the plasma membrane, where it colocalizes with the marker H-36, and also some colocalization with recycling and early endocytic compartments [30]. Additionally, Polycystin-2 had been confirmed to be localized at the plasma membrane and some cytoplasmic puncta as visualized using a GFP fusion protein [31] and with an anti-Polycystin-2 antibody [34]. The role of Polycystin-2 in *Dictyostelium* appears to be associated with calcium regulation. Polycystin-2 knockout strains exhibit defects in Ca^2+^-induced lysosomal exocytosis, indicating its involvement in cytosolic calcium release, which regulates this process [30]. Additionally, Polycystin-2 may be associated with calcium signalling related to rheotaxis [30], although the reproducibility of this phenotype has been questioned [31,35]. Polycystin-2 knockout strains show impaired cytosolic calcium influx in response to ATP and ADP, while still retaining responses to cAMP and folic acid [31]. The regulatory mechanisms and potential interactions of Polycystin-2, such as its interaction with the putative PC1 homologue encoded in the *Dictyostelium* genome (dictyBase gene ID: DDB_G0289409), remain unknown and require further investigation. Clearly, additional studies are necessary to elucidate the role of Polycystin-2 in calcium signalling and its mode of action.

Here we have conducted a comprehensive analysis of the role of Polycystin-2 by examining phenotypic readouts in strains overexpressing the protein, or in knockdown strains (KD) with reduced levels of the protein resulting from antisense RNA inhibition (asRNA). The focus was on characterizing the contribution of Polycystin-2 to chemotactic calcium signalling in responses to the stimuli cAMP (in aggregation-competent cells) and folic acid (in vegetative, growth-phase cells). The use of knockdown and overexpression systems allowed us to correlate phenotypic readouts with the copy number of the constructs. This approach involved multiple, independently isolated mutant strains with different numbers of copies of the knockdown and overexpression constructs. It enables regression analysis of phenotype versus copy number and reduces the likelihood of unknown off-target genetic events influencing the outcomes. Consequently, this approach increased the sensitivity to detect even subtle phenotypic abnormalities that may not be apparent in the analysis of KO strains. The results show that overexpression of Polycystin-2 enhances cytosolic calcium responses to both cAMP and folic acid while knockdown reduces these responses, implicating Polycystin-2-dependent calcium fluxes in contributing to these responses. These alterations in calcium signalling also have downstream effects on other calcium-sensitive processes, including growth, endocytosis, cell differentiation and spore viability. 

## 2. Materials and Methods

### 2.1. Gene Cloning and Sequence Analysis

Vector construction and cloning procedures were conducted as described previously [36,37]. Gene fragments for cloning were amplified using gene-specific primers incorporating restriction enzyme cut sites at the 5′end to facilitate cloning. The 2717 bp *Dictyostelium* homologue of Polycystin-2 (*pkd2*), identified by Wilczynska et al., (2005) [29] (dictyBase gene ID. DDB_G0272999), was amplified in two sections from AX2 genomic DNA with primers POLYF 5′-CGCGGATCCATCGATATGAATACATTAAAGAGGACAGT-3′ and PolyMR 5′-CATATATAATT*GAATTC*TATAAGTTC-3′ for the 5′ section and PolyMF 5′-GAACTTATA*GAATTC*AATTATATATG-3′ and POLYR 5′-CGCGGATCCCTCGAG TTAAGGTGTATTAGTACCACCA-3′ for the 3′ section. The full-length gDNA was cloned in two sections into the bacterial vector pZErO-2 (Invitrogen, Carlsbad, CA, USA) using the restriction sites indicated by underlining and utilization of a resident *Eco*R1 restriction site (indicated by italics) within the gene sequence, which facilitated seamless ligation of the 5′ and 3′ gene fragments (pPROF635). The gene was then subcloned for overexpression into the vector pA15GFP, replacing the resident green fluorescent protein gene (pPROF636). Plasmid constructs for expression of *pkd2* antisense RNA were created by amplifying a 787bp fragment of the gene, via reverse transcription-PCR with the primers POLYF and PolyMR. Template RNA was extracted from vegetative AX2 cells with TRIzol^®^ (Invitrogen). The gene products were then cloned into pZErO™-2 and subcloned in the antisense orientation into the *Dictyostelium* expression vector pDNeo2 (pPROF646).

Sequencing was conducted by The Australian Genome Research Facility (AGRF) (http://www.agrf.org.au). Sequences we analysed using the software Basic Local Alignment Search Tool (BLAST) version 2.2.23 (National Center for Biotechnology Information (NCBI), Bethesda, MD, USA, accessed on 12 February 2009), and ClustalW version 2 (European Molecular Biology Laboratory—European Bioinformatics Institute, Cambridge, UK, accessed on 6 August 2009). These software were accessed through ExPASy (http://www.expasy.org, SIB Swiss Institute of Bioinformatics, Lausanne, Switzerland). Database searches were performed through dictyBase (http://www.dictybase.org, accessed on 30 August 2005).

### 2.2. Dictyostelium Transformation

Strains were created by the transformation of the parental strain AX2 as described previously [36,37,38]. Each strain (assigned HPF codes) carried different numbers of copies of the following constructs: (1) pPROF120 and pPROF646 (antisense HPF642-644, HPF831, HPF833, HPF835-837), (2) pPROF120 and pPROF648 (sense HFP838 & HPF848) and (3) pPROF120 and pPROF636 (Overexpression HPF839-841, HFP644-646, HPF651-653). The transformation was carried out via the Ca(PO_4_)_2_/DNA coprecipitation method by cotransformation with both plasmids [39]. Isolated single colonies were selected from lawns of *Micrococcus luteus* grown on Standard Medium (SM) agar containing 25 µg/mL G418 [40]. 

### 2.3. Strains and Culture Conditions

*Dictyostelium* cultures were subsequently maintained on *Enterobacter aerogenes* lawns prepared on SM agar, and axenically in HL5 medium [41]. To remove any possible effects of the antibiotics on phenotypic outputs, strains were subcultured at least one time into HL5 without antibiotics and grown for 24–48 h prior to use in experiments. Cells were harvested by centrifugation at 500× *g* for 5 min, unless otherwise stated.

### 2.4. Estimation of Plasmid Copy Number Using Quantitative Real-Time PCR

The copy number of each plasmid inserted into the genome of the *Dictyostelium* strains was determined by quantitative real-time PCR (qRT-PCR) as described previously [41]. The primers used were: Polycystin-2 (103 bp amplicon): Forward primer 5′-CGCGGATCCAAGCTTATGAATACATTAAAGAGGACAGT-3′, Reverse primer 5′-CAACTGCTGCCAATAATGTTGC-3′. Filamin (100 bp amplicon of *abpC* gene): Forward primer 5′-CCCTCAATGATGAAGCC-3′, Reverse primer 5′-CCATCTAAACCTGGACC-3′.

### 2.5. Quantification of mRNA Expression Levels

Semiquantitative reverse transcription PCR was used to quantitate expression levels of *pkd2*. RNA was extracted from 10^7^ vegetative cells or cells harvested from water agar every 2 h over 24 h for the developmental time course using TRIzol^®^ (Invitrogen) and treated with RQ1 DNase (Promega, Madison, WI, USA) according to the manufacturer’s instructions. The iScript™ One-Step RT-PCR Kit with SYBR^®^ Green (BioRad, Hercules, CA, USA) was used for amplicon detection using the primers outlined in Section 2.4. An RNA template of 50–100 ng was added to the total reaction mixture of 50 μL; cDNA synthesis was performed at 50 °C for 10 min, and the PCR cycling and detection used 45 cycles of denaturation at 95 °C for 30 s, annealing at 58 °C for 30 s and extension and data collection at 72 °C. Expression was normalized against the *abpC* mRNA levels to adjust for loading and then measured relative to AX2 controls. 

### 2.6. Plaque Expansion Rates on Bacterial Lawns and Determination of Generation Times of Axenically Growing Cultures

Growth experiments were conducted as described previously [41]. The expansion rates of *Dictyostelium* plaques on *E. coli* B2 lawns were assessed by measuring plaque diameter every 8–10 h for 100 h. Generation times of axenic *Dictyostelium* cultures in HL5 were determined by hemocytometer cell counts at 8–12 h intervals over 72 h. The recorded values were analysed by linear regression using the “R” environment for statistical computing and graphics (http://www.R-project.org, accessed on 28 June 2006) to determine the plaque expansion rate (mm/h) on bacterial lawns, or the generation time (h) from the exponential growth curve in shaken axenic culture.

### 2.7. Endocytosis Assays

Phagocytosis levels in *D. discoideum* strains were assessed by quantifying the uptake of an *E. coli* strain that expressed the fluorescent protein DsRed [42], following the method described previously [41]. Meanwhile, the rates of macropinocytosis were evaluated by measuring the uptake of fluorescein isothiocyanate (FITC)-dextran (Sigma-Aldrich, average molecular weight 70 kDa, St. Louis, MO, USA), also following the procedure described [41].

### 2.8. Morphology

To observe mature fruiting body morphology, cells were allowed to develop on lawns of *Enterobacter* as previously described [37]. After 24 h, images were taken using a Moticam 2300 camera attached to an Olympus SZ61 stereomicroscope from above and from the side of an excised piece of agar containing the fruiting body.

### 2.9. Quantification of LysoSensor™ Blue DND-167 Staining in Cells

*Dictyostelium* cells were grown in HL5 to a density of 1–3 × 10^6^ cells mL^−1^ and then 1 × 10^6^ cells were harvested (1500× *g* for 2 min) and resuspended in 1 mL of Lo-Flo HL5 (3.85 gL^−1^ glucose, 1.78 gL^−1^ Proteose peptone, 0.45 gL^−1^ yeast extract, 0.485 gL^−1^ KH_2_PO_4_ and 1.2 gL^−1^ Na_2_HPO_4_·12H_2_O; filter sterile). Aliquots of 100 µL of the cell suspension were seeded into 4 wells of a black 96-well microplate. To two wells, 100 µL of Lo-Flo HL5 was added as background controls, and to the other two wells, 100 µL of 1µM LysoSensor^TM^ Blue DND-167 (Invitrogen) was added. The plate was incubated in the dark for 30 min at 21 °C, and the fluorescence was recorded on a CLARIOstar plate reader (Ex.373 Em.425) The background fluorescence of the undyed cells was subtracted from the fluorescence recorded from the Lysosensor Blue™ stained cells. 

### 2.10. Determination of Spore Viability

*Dictyostelium* strains were grown to a density of 1–2 × 10^6^ cells mL^−1^ in the HL5 medium. 1 × 10^7^ cells were harvested (500× *g* for 5 min) and washed 2× in PBS. The cell pellet was resuspended in 50 μL PBS and pipetted in a 1 cm^2^ area on non-nutrient agar and incubated at 21 °C until fruiting bodies had formed. The fruiting bodies were scraped from the surface using a sterile tip, resuspended in 1 mL PBS and washed 2× in PBS (1000× *g* 2 min), then resuspended in PBS (0.1% *v*/*v* NP40) and incubated at 42 °C for 30 min to kill amoebae and activate spores [43,44]. The spores were washed in 2× in PBS and counted using a hemocytometer. Duplicate plates of 300 and 100 spores were plated onto SM agar by mixing with a thick slurry of *E. aerogenes* and evenly spread onto the agar. Plates were incubated at 21 °C for ~3 days until spores had germinated and plaques had formed. Each plaque was assumed to have arisen from an individual spore. The number of plaques was counted, and the spore viability was calculated as the percentage of plated spores that produced plaques.

### 2.11. Calcium Experiments

Expression of the Ca^2+^ sensitive luminescent protein apoaequorin, in each *Dictyostelium*, strain was used to measure cytosolic Ca^2+^ concentration ([Ca^2+^]_c_) as described previously [41]. Briefly, axenic cultures in HL5 were grown to 1–2 × 10^6^ cells mL^−1^, with 1 × 10^8^ cells harvested. Cells were either resuspended in HL5 (2 × 10^7^ cells ml^−1^) or to initiate aggregation competence washed twice in 20 mL MES-DB (10 mM MES/NaOH, pH 6.2, 10 mM KCl, 0.25 mM CaCl_2_), and resuspended to 2 × 10^7^ cells ml^−1^ in MES-DB. Cells were loaded with 5 µM coelenterazine-*h* (Invitrogen) and incubated at 21 °C and 150 rpm for 4 h (for experiments using vegetative cells) or 6–7 h (for experiments with aggregation competent cells). Determination of the consumption of aequorin and measurements of in vivo Ca^2+^ concentrations were conducted in a Lumitran^®^ model L-3000 photometer (New Brunswick Scientific, Edison, NJ, U.S.A ). Luminescence signals were recorded from 5 mL cell suspensions stirred at 100 rpm stimulated with chemoattractant (final concentration 1 µM). A model PCI-20428 multifunction I/O data acquisition board (Intelligent Instruments Pty. Ltd., Southampton, Hampshire, England) was used to capture the signal, which was then converted into values of [Ca^2+^]_c_ on a PC using purpose-designed software developed by (Prof. P. R. Fisher) written in the Intelligent Instruments graphical programming language for the PCI-20428. Subsequent statistical and graphical analyses were performed in the WinStat Excel add-on and in the ‘R’ programming language for data analysis and graphical display.

### 2.12. Statistical Analysis

Data analysis was conducted using Microsoft Excel and the ‘R’ programming language. Independent *t*-tests or One-Way ANOVA, with pairwise comparisons made by the Least-Squares Difference method, were used to compare means. Pearson product-moment, Spearman’s rank and Kendall’s rank correlation coefficients were used to determine significant correlations. Regression analysis was performed as indicated in the figure legends.

## 3. Results

### 3.1. Developmental Expression of pkd2

Upon starvation, *Dictyostelium* cells begin a developmental program that involves various stages, leading to multicellular differentiation and ending in the formation of mature fruiting bodies [45]. Gene expression begins to change immediately upon starvation and can vary over the developmental period [46]. To investigate the expression of *pkd2* during development, AX2 cells were plated onto water agar and starved for 24 h, during which time RNA samples were harvested every 2 h to capture the different developmental stages. The relative expression of *pkd2* was assessed using semiquantitative RT-PCR. *pkd2* showed continuous expression throughout development, increasing swiftly to a peak within the initial 2 h of early development in reaction to the initiation of starvation. Subsequently, its expression declined between 2 and 8 h, before experiencing another rise during aggregation, reaching its highest level at 10 h. Throughout multicellular development, expression levels remained elevated (Figure 1A). Our expression data show a similar trend to the RNAseq-derived mRNA expression time course of *pkd2* in AX4 developed on KK2 filter paper as determined by Parikh A et al. [47], represented in Figure 1B from data retrieved from dictyexpress.org [48,49]. Our results are also broadly similar to the mRNA developmental time course previously published [31], except in that case, the initial peak at 2 h in expression is higher than that of expression later in development. These differences in detail may be due to differences in the expression assays or in the developmental conditions. Combined, these expression patterns suggest roles for Polycystin-2-dependent calcium signalling during early starvation-induced development, and also during the lifecycle multicellular stages.

### 3.2. Genetic Manipulation of pkd2 Expression by Transformation of D. discoideum with Plasmid Constructs

*Dictyostelium* strains were generated by transforming them to express constructs that either suppress mRNA expression through antisense mRNA inhibition- knockdown (KD) strains or enhance mRNA expression–overexpression strains. Plasmid integration takes place randomly in the genome through a process known as recombination and rolling circle replication. The outcome of this process produces individual strains with varying copy numbers of integrated plasmids, resulting in differing levels of mRNA expression [50]. To determine the number of copies integrated into the genome of each transformant, quantitative real-time PCR was used, and semiquantitative real-time RT-PCR was used to determine the subsequent mRNA expression levels. A regression analysis was then carried out to determine any relationship. As anticipated, antisense inhibition led to a reduction in mRNA expression, overexpression increased expression and there was a significant correlation between copy number and mRNA expression (Figure 2). This allowed us to create a *pkd2* expression index for phenotypic analysis, following the previously published convention of using negative values (the antisense inhibition construct copy number) and positive values (the overexpression construct copy number) [38]. This allowed gene dose–response relationships to be determined and revealed correlations between the construct copy number and the severity of phenotypes. 

### 3.3. Polycystin-2 Contributes to Chemotactic Calcium Responses in Dictyostelium

To further investigate the role of Polycystin-2 in Ca^2+^ signalling, we used our Polycystin-2 knockdown and overexpressing strains to measure changes in cytosolic calcium concentrations in vegetative cells stimulated with folic acid, as well as in aggregation-competent cells stimulated with cAMP. Real-time cytosolic Ca^2+^ responses were measured using these strains, which also expressed a luminescent recombinant Ca^2+^-sensitive protein, apoaequorin. The cytosolic Ca^2+^ responses in cells stimulated with folic acid and cAMP were already well-characterized in the parental strain AX2 expressing aequorin (HPF401) [51]. This method provides sensitive and accurate readings with a temporal resolution of 20 ms and is sensitive enough to detect changes in [Ca^2+^]_c_ of 2 to 3 nM [29]. Representative real-time recordings of [Ca^2+^]_c_ were measured in three strains—the control strain expressing aequorin (HPF401), the Polycystin-2 knockdown strain (HPF644) and the Polycystin-2 overexpression strain (HPF839)—which are shown in Figure 3. Vegetative cells were stimulated with 1 μM folic acid (Figure 3 upper panel) and optimally starved cells were stimulated with 1 μM cAMP (Figure 3 lower panel). Larger and smaller response magnitudes, compared to control (HPF401), were observed in overexpression and KD strains, respectively, for both folic acid and cAMP stimulation. In vegetative cells, response magnitudes decreased by an average of 45.5% in KD strains and increased by 48% in overexpression strains compared to the control average. In differentiated cells, response magnitudes were decreased by an average of 35.8% in KD strains and increased by 42.8% in overexpression strains compared to the control average. A regression analysis was then conducted to analyse the mean response magnitudes from five different KD strains and five different overexpression strains carrying different plasmid copy numbers. The magnitude of the Ca^2+^ responses, to both chemoattractants, was positively correlated with the *pkd2* expression index (Figure 4A,B). The observation of similar copy number-dependent phenotypes in multiple, independently isolated clonal cell lines excludes the possibility that the altered Ca^2+^ responses might be due to additional, random and unknown genetic events elsewhere in the genomes of the mutant strains. Interestingly, a previous study reported that Polycystin-2 KO cells still retain their cytosolic calcium responses to folic acid and cAMP measured in single cells expressing a genetically encoded cytoplasmic calcium sensor [31]. This means that Polycystin-2 cannot be the only calcium channel involved in facilitating this calcium influx. However, this does not exclude the possibility that Polycystin-2 contributes to the calcium influx, and our results support its role in contributing to cytosolic Ca^2+^ release into the cytoplasm.

#### 3.3.1. Analysis of the Kinetics of Ca^2+^ Responses to Chemoattractant

The kinetics of these chemotactic Ca^2+^ responses are known to be partly dependent on the response magnitude itself. Larger responses start earlier and peak earlier due to the faster increase in cytosolic Ca^2+^, and the maximum rise and fall rates of Ca^2+^ flux into and out of the cytoplasm are correlated [29]. This confirms that the responses are autocatalytic during the rising phase when calcium is entering the cytosol, implying Ca^2+^-induced Ca^2+^ entry into the cytosol with more channels opening earlier and allowing more rapid increases in cytosolic Ca^2+^. The responses are also self-limiting in the falling phase of the response, with larger responses being terminated more quickly because the Ca^2+^ itself homeostatically activates the mechanisms that clear Ca^2+^ from the cytoplasm, such as closure of Ca^2+^ channels or activation of Ca^2+^ pumps [29].

To explore these relationships further, we measured the maximum rise and fall rates of cytosolic Ca^2+^ in *pkd2* mutants and observed the same relationship in response kinetics to both cAMP and folic acid (Figure 5A). This shows that the Ca^2+^-regulated Ca^2+^ channels and pumps are the same for the two attractants since responses to them exhibit the same kinetic properties. We also plotted the relationship between the response onset time and response magnitude. The relationship is negative for Polycystin-2 overexpression, KD and control strains (Figure 5B), and in this respect, is similar to the relationship previously reported for wild-type, calnexin-deficient and calreticulin-deficient strains [29]. Calnexin and calreticulin are avid Ca^2+^-binding proteins in the endoplasmic reticulum that contribute to the sequestration of Ca^2+^ in the lumen of the ER. The data for all three of those strains fell onto the same regression line, with the strains differing only in their position on the line. This was in keeping with differences in the steepness of the free Ca^2+^ gradients resulting from the loss of Ca^2^-sequestering capacity in the ER. In this present work, however, each set of strains falls on a different line in what appears to be a family of similar log-linear regressions. In overexpression strains, the response magnitude is greater than the controls by a constant amount (regression intercept increased but slope unchanged) at all response times. Conversely, the response magnitude is reduced by a constant amount, independent of response times, compared to controls (the intercept is lower but the slope is unchanged) (Figure 5B). The simplest explanation is that Polycystin-2 is a significant contributor to the Ca^2+^ responses at all response times and magnitudes, but that its expression levels do not affect the mechanism by which response time and magnitude are coupled, e.g., in the form of Ca^2+^-induced Ca^2+^ release. This is consistent with the hypothesis that Polycystin-2 in *Dictyostelium* is not itself Ca^2+^-regulated, as predicted by its lack of sequence motifs for known Ca^2+^- or Ca^2+^-calmodulin-binding sites [29].

#### 3.3.2. Resting Ca^2+^ Levels Are Altered by Changing *pkd2* Expression

The ability of cells to maintain appropriate [Ca^2+^]_c_ is crucial for tight regulation of the many Ca^2+^-regulated cellular processes. Assessment of the basal [Ca^2+^]_c_ in the *pkd2* mutant strains revealed no significant correlation between copy number and resting Ca^2+^ levels, therefore data from all transformants within each group were pooled to calculate the mean (Figure 6A,B). The results showed significant reductions in the basal [Ca^2+^]_c_ of KD strains compared to the control, both in the vegetative (*p* < 0.01) and developed states (*p* < 0.05). However, there was no significant difference in the basal [Ca^2+^]_c_ of *pkd2-*overexpressing strains compared to control. These results show that Polycystin-2 not only contributes to cytosolic Ca^2+^ transients in chemoattractant responses but also contributes to basal Ca^2+^ homeostasis in unstimulated cells. The magnitude of this contribution is smaller when Polycystin-2 levels are reduced but is already at or close to its maximum at wild-type expression levels and unresponsive to overexpression.

### 3.4. pkd2 Expression Levels Affect Fruiting Body Morphologies and Spore Viability

In *Dictyostelium*, starvation triggers multicellular development, leading to the formation of mature fruiting bodies consisting of spore heads, stalks and basal discs (Figure 7A shows a wild-type AX2 fruiting body). Cells that differentiate into stalk and disk cells undergo a form of autophagic cell death [52,53], and the viable spores in the sorus can disperse and grow into amoebae when conditions improve. Aberrant Ca^2+^ signalling affects downstream processes, including multicellular morphogenesis [54] (Schaap et al. (1996)). Prolonged elevation of cytosolic Ca^2+^ levels (as occurs during multicellular development) mediates the induction of prestalk gene *ecmB* and stalk cell differentiation by the *Dictyostelium* morphogen DIF (Differentiation Inducing Factor) [55]. In the present study, when *pkd2* expression was knocked down, transformants developed into fruiting bodies with very thin fragile stalks that were often unable to support the sorus (Figure 7B). Conversely, when we overexpressed *pkd2,* cytosolic Ca^2+^ signalling was enhanced and the transformants developed fruiting bodies with very thick stalks, presumably as a consequence of excess stalk cell differentiation (Figure 7C). 

*Dictyostelium* spores remain dormant until activated for germination, a process involving Ca^2+^ signalling [43,56]. Upon spore activation, Ca^2+^ is released through IP_3_-dependent Ca^2+^ release and subsequent CaM-dependent efflux. During amoebal emergence, Ca^2+^ uptake is essential for emergence-specific protein production [57]. We evaluated spore viability in Polycystin-2 mutant strains by harvesting and heat-activating spores, and then plating them on *Enterobacter* lawns. Results showed significantly decreased spore viability in overexpression strains compared to KD strains (Figure 8). While we expected that the increased calcium signalling in the Polycystin-2 overexpression strains may have increased spore viability, one possibility is that elevated Ca^2+^ levels/responses caused by Polycystin-2 overexpression caused increased premature spore germination in the sorus of the fruiting body, so a higher proportion is not viable on the plated lawns.

### 3.5. Polycystin-2 Positively Regulates Growth Rates and Nutrient Uptake via Phagocytosis and Pinocytosis

Calcium signalling plays a vital role in endocytic processes including pinocytosis and phagocytosis. It controls vesicle formation, membrane trafficking and cytoskeletal dynamics involved in endocytosis [58]. It also regulates proteins and enzymes responsible for vesicle formation and fusion [59,60]. Similarly, calcium signalling regulates various steps of phagocytosis, including particle recognition, engulfment and vesicle formation. It Influences cytoskeletal rearrangements, receptor activation and signalling molecule recruitment during phagocytosis [61]. Therefore, it was crucial to assess these processes in our Polycystin-2 mutant strains to identify any abnormal phenotypes. We evaluated pinocytosis by measuring the uptake of FITC-dextran over a 70 min period. To determine the rates of phagocytosis, we measured the uptake of fluorescently labelled live *E. coli* cells over a 30 min duration. The uptake of DsRed-labelled *E. coli* and FITC-dextran HL5 both showed a positive correlation with the Polycystin-2 expression index and are therefore positively regulated by Polycystin-2, as illustrated in Figure 9A,B. 

We proceeded to investigate whether the observed heightened nutrient uptake in our mutants translated into improved growth rates. Other studies on Polycystin-2 KO cells have yielded conflicting results regarding growth rates [31,33]. Our investigation revealed that the increased rates of pinocytosis and phagocytosis resulting from increased *pkd2* expression corresponded with faster axenic growth and plaque expansion rates on bacterial lawns (Figure 10A,B). These findings provide compelling evidence that the altered rates of axenic growth and plaque expansion in Polycystin-2 transformants are a result of altered rates of pinocytosis and phagocytosis.

### 3.6. Polycystin-2 Expression Affects the Endolysosomal Vesicles

Our results indicate that Polycystin-2 expression can influence endocytic processes and autophagic cell death. Given that Polycystin-2 expression has been detected in recycling and early endocytic compartments and is involved in controlling lysosomal exocytosis [30], we decided to further assess how Polycystin-2 expression levels affect the vesicles of the endolysosomal compartments. We stained vegetative cells with LysoSensor^TM^ Blue DND-167, a fluorescent dye with a pKa of ~5.1, that accumulates in acidic organelles within cells, particularly lysosomes, and becomes more fluorescent in acidic environments. The recorded fluorescence increased as Polycystin-2 expression increased, indicating that Polycystin-2 either increases lysosomal mass or, because LysoSensor™ Blue DND-167 fluorescence increases in acidic environments, increases the acidification (decreased pH) in these compartments (Figure 11). 

## 4. Discussion

In this study, Polycystin-2 function in *Dictyostelium* was investigated by knocking down and overexpressing *pkd2* mRNA and assaying a variety of phenotypic readouts in transformants with different levels of expression. In the absence of a specific anti-Polycystin-2 antibody, we were unable to measure the levels of the protein itself. However, the fact that the phenotypic outcomes were correlated with mRNA expression levels supports the inference that the protein levels were also correlated with the copy numbers of the antisense and overexpression constructs. Polycystin-2′s role in chemotactic Ca^2+^ signalling was assessed using in vivo expression of the Ca^2+^-sensitive luminescent protein, and aequorin and chemotactic Ca^2+^ responses in these transformants were characterized. As *Dictyostelium* is an excellent model organism for the analysis of the pathophysiology behind many diseases, its potential use as a model for ADPKD was assessed.

Polycystin-2 in the *Dictyostelium* parental strains DH1-10 and AX2 cells localizes at the plasma membrane [30,31,34], which suggests that it may indeed be a Ca^2+^ channel facilitating extracellular Ca^2+^ influx. To investigate whether Polycystin-2 can contribute to calcium influx, we measured cytosolic Ca^2+^ responses to chemoattractants in Polycystin-2 knockdown (KD) and overexpression transformants. If Polycystin-2 was capable of contributing to the Ca^2+^ responses, we expected that overexpressing and knocking down the expression of the protein would increase and decrease the Ca^2+^ responses, respectively. Indeed, analysis of cytosolic Ca^2+^ responses revealed that they were significantly altered. Overexpressing Polycystin-2 significantly amplified the magnitude of the Ca^2+^ responses to 1µM cAMP and 1µM folic acid compared to wild type. Furthermore, in KD transformants, the Ca^2+^ responses were reduced, as were the basal cytosolic Ca^2+^ levels. These results imply not only that Polycystin-2 contributes to the responses but also to maintaining the cell’s resting Ca^2+^ levels. 

Analysis of the maximum rise and fall rates of cytosolic Ca^2+^ in Polycystin-2 mutants revealed the same relationship in response kinetics to both cAMP and folic acid and shows that the Ca^2+^-regulated Ca^2+^ channels and pumps are the same for the two attractants since responses to them exhibit the same kinetic properties. The relationship between the response onset time and response magnitude was negative for Polycystin-2 overexpression, KD and control strains, and in this respect, is similar to the relationship previously reported for wild-type, calnexin-deficient and calreticulin-deficient strains [29]. However, in this work, each set of strains falls on a different line so that, in overexpression strains, the response magnitude is greater than the controls by a constant amount at all response times. Conversely, in the KD strains, the response magnitude is reduced by a constant amount, independent of response times, compared to controls. The simplest explanation is that Polycystin-2 is itself a significant contributor to the Ca^2+^ responses at all response times and magnitudes, but that the mechanism by which the response time and magnitude are coupled is unchanged, e,g., in the form of Ca^2+^-induced Ca^2+^ release. This makes sense if Polycystin-2′s contribution to Ca^2+^ responses is not Ca^2+^-regulated, as expected since the Polycystin-2 sequence does not contain recognizable Ca^2+^- or Ca^2+^-calmodulin-binding sites [29]. These results provide compelling evidence for Polycystin-2 as a Ca^2+^ channel. However, Polycystin-2 cannot be the only plasma membrane calcium channel involved in chemotactic calcium responses because Polycystin-2 KO cells still retain their calcium responses when stimulated with cAMP and folic acid [31]. Additionally, Polycystin-2 KO cells are still able to undergo chemotaxis towards a source of folic acid [30]. This does not exclude the possibility of Polycystin-2 contributing to these Ca^2+^ responses given the overlapping and redundant functions of many calcium channels and pumps. Our results therefore show that Polycystin-2 does contribute to calcium influx during calcium responses to cAMP and folic acid even though it is not essential for them.

If Polycystin-2 is not directly gated by either of the chemoattractants used in this work (folate and cAMP), both of which interact with known G-protein-coupled receptors, and is not responsive to intracellular Ca^2+^ levels either, what might be the signal that controls its activity during responses to these chemoattractants? The answer might lie in Polycystin-2′s ability to interact with other plasma membrane proteins, as is the case with its mammalian counterpart. A candidate calcium channel that may reside at the plasma membrane to facilitate calcium influx is IplA, a putative calcium channel in *Dictyostelium* resembling InsP3 receptors [62]. Indeed, a double KO of Polycystin-2/IplA completely lacks chemotactic calcium responses [31]. This prompts the question of whether there exists an interaction between the two proteins, and when they are co-localized within the same sub-cellular compartment, whether they jointly contribute to the cytosolic Ca^2+^ response. Indeed, in other organisms, InsP3 receptors and Polycystin-2 can directly interact to form a functional complex. InsP3 receptors have been found to gate PC2 channel activity by interactions between the two channels through acidic residues in the N-terminal ligand-binding domain of the InsP3 receptor and the C-terminal ER retention signal of PC2 [13,15]. In *Dictyostelium*, coimmunoprecipitations of IplA and Polycystin-2 would uncover whether the channels also interact. Intriguingly, Polycystin-2 was shown by Traynor and Kay (2017) [31] to be essential for Ca^2+^ responses to extracellular ATP, suggesting that it may interact with purinergic receptors [63]. Another potential interaction partner of Polycystin-2 is the putative PC1 homologue that is encoded in the *Dictyostelium* genome (dictyBase gene ID: DDB_G0289409). Characterization of this protein is needed to determine its involvement in Ca^2+^ signalling and to reveal whether it functions as a PC1-like protein interacting with Polycystin-2.

Aberrant Ca^2+^ signalling can have a profound impact on numerous Ca^2+^-dependent cellular processes. However, previous reports regarding the phenotypic effects of Polycystin-2 knockout have yielded conflicting results among different laboratories. While some knockout strains exhibited rheotaxis defects [30], others did not [31,35]. Similarly, growth defects were observed in some KO strains [33], but not in others [31]. These varying phenotypes may be subtle and not easily detectable in a binary comparison of KO versus wild-type cells.

In order to uncover more nuanced phenotypes, our experiments employed KD and overexpression, an approach that allows the correlation of phenotypic severity with an expression index in multiple, independently isolated strains. The results revealed that Polycystin-2 positively regulates growth rates, as well as the nutrient uptake processes of macropinocytosis and phagocytosis. The defects observed in the overexpression strains were more pronounced, but the defects in the KD strains were mild, perhaps explaining some of the undetected phenotypes in Polycystin-2 KO strains from other studies. In our study, a strong correlation between the Polycystin-2 expression index and the phenotypic defects highlighted the regulatory role of Polycystin-2 in these processes. Polycystin-2 likely modulates Ca^2+^ signalling, as many steps of the endocytic pathway in mammalian cells are known to be Ca^2+^-regulated and the vesicle sources of stored calcium [64,65].

In favour of a regulatory role of Polycystin-2-dependent Ca^2+^ signalling in endosomal trafficking, KO cells had reduced Ca^2+^-stimulated lysosome exocytosis [30]. Other studies suggest that Ca^2+^ signalling is involved in the endocytic process in *Dictyostelium*. For example, KO mutants of fAR1 (the folate receptor in proliferating cells), upon binding to bacteria, have a greatly reduced ability to induce actin polymerization or produce phagocytic cups, potentially due to the absence of fAR1-associated chemotactic signalling which involves Ca^2+^ [51,66]. The calcium-regulated actin-bundling protein, fibrin, is associated with both the macrophagosome and macropinosome during formation [67]. Furthermore, the Ca^2+^-binding proteins calnexin and calreticulin are essential for the formation of the phagocytic cup [68]. Given its location at the plasma, recycling endosome and newly formed endosome membranes [30], Polycystin-2 is well-situated to contribute to the regulation of membrane fusion/fission by Ca^2+^ signalling in these early events. 

An indication that Polycystin-2 regulates other processes along the endolysosomal pathway is evident, as cells stained with Lysosensor Blue™ displayed either decreased or increased fluorescence in our Polycystin-2 KD or overexpression strains, respectively. These changes in fluorescence could represent decreased or increased lysosomal mass, respectively, or changes in the pH of the lysosomal compartments. The luminal calcium concentration of endocytic vesicles has been linked to acidification [69], so the presence of Polycystin-2 in the membrane of early endosomes may help to control calcium homeostasis during acidification. We saw similar abnormalities in lysosomal mass or acidification in *Dictyostelium* strains when knocking down or overexpressing the calcium channel, mucolipin, another TRP channel that localizes to endocytic compartments [41]. Changes in lysosomal mass could signify abnormalities in progression through the interconnected endocytic and autophagic pathways [70], both of which are, in part, regulated by calcium signalling [71,72]. For instance, if lysosomes fail to fuse properly with autophagosomes during the autophagy process, the undegraded contents of autophagosomes can accumulate in the cell, along with an increased number of lysosomes.

Indeed, we did observe indications of abnormalities in autophagic cell death in our Polycystin-2 strains. The autophagic cell death pathway in *Dictyostelium* plays an important role in multicellular development leading to the differentiation of stalk cells, a process that is mediated by DIF, which induces prolonged, elevated intracellular Ca^2+^ levels and induces expression of the prestalk cell gene *ecmB* [55]. Our Polycystin-2 KD and overexpression strains exhibited abnormal stalk cell differentiation—the overexpression strains developed short thick stalks while the knockdown strains had thin, fragile stalks reminiscent of the delicate fruiting bodies frequently observed to collapse in Polycystin-2 KO cells [31]. This implicates Polycystin-2 mediated calcium signalling in stalk cell differentiation; however, as Polycystin-2 KO cells still retain their calcium responses to DIF stimulation [31], other calcium channels must also be involved. The differentiation of stalk and disk cells represents a form of autophagic cell death in *Dictyostelium* [52,53], and shorter thicker stalks have been associated with more cells entering the stalk differentiation and autophagic cell death pathway [38]. These phenotypes may reflect aberrant Ca^2+^ signalling and suggest that precise control of Ca^2+^ release by Polycystin-2 is necessary to maintain normal stalk/spore proportions within the population. This process is known to be Ca^2+^-regulated as cell-type specific differentiation into prespore and prestalk cells is controlled partly by Ca^2+^, such that higher Ca^2+^ levels tend to direct cells to the prestalk pathway and lower Ca^2+^ direct cells to the prespore pathway [54,55,73,74,75,76].

Defective autophagy is reported in ADPKD [77], and a role for PC2-dependent Ca^2+^ signalling in autophagy is emerging [78,79,80]. The overexpression of PC2 has been shown to induce autophagy, which was attenuated by the Ca^2+^ chelator BAPTA-AM, suggesting the regulation of autophagy by PC2 is contingent on intracellular calcium control [78]. In human cardiomyocytes derived from embryonic stem cells, PC2 knockdown reduces caffeine-induced cytosolic Ca^2+^ rise and autophagic flux during glucose starvation, affecting the activation of the key central signalling complexes AMP-activated protein kinase (AMPK) and mammalian target of rapamycin (mTOR) [81]. PC2 activity is also essential for inducing autophagy under hyperosmotic stress in HeLa and HCT116 cells via the mTOR pathway [80]. These parallels with our findings confirm that *Dictyostelium* is an ideal model for studying autophagy and autophagic cell death [82,83], and given the extensive knowledge available AMPK and DdTOR signalling in *Dictyostelium* [38,84,85,86,87], this model system offers invaluable opportunities to investigate Polycystin-2-regulated autophagy and the underlying signalling pathways.

## 5. Conclusions

This study has provided evidence that Polycystin-2 contributes to cytosolic Ca^2+^ signalling in response to chemotactic stimuli in *Dictyostelium*. Further, Polycystin-2 dependent Ca^2+^-signalling must be tightly controlled to regulate downstream Ca^2+^-sensitive processes such as endocytosis, growth and autophagic cell death. Importantly, this study has also highlighted the potential to use *Dictyostelium* as a model to study the underlying molecular functioning behind ADPKD.

## Figures and Tables

**Figure 1 cells-13-00610-f001:**
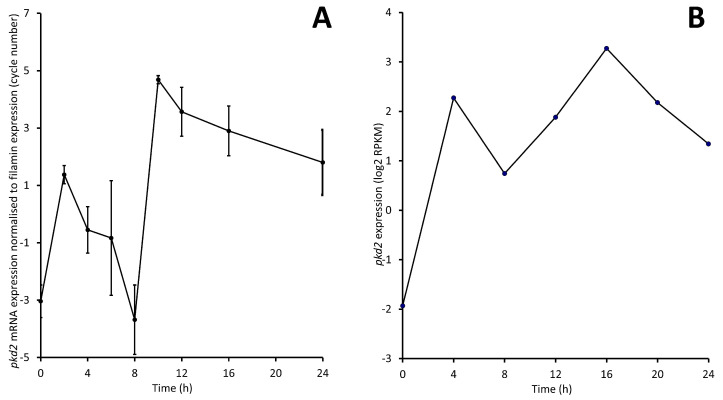
Developmental changes in *pkd2* mRNA expression in *Dictyostelium* AX2 and AX4. (**A**) Relative *pkd2* expression was measured by semiquantitative RT-PCR in AX2 over 24 h. *pkd2* expression (cycle number) was normalized to *abpC* (filamin gene) expression. Cells were plated and allowed to develop on water agar, and every two hours cells were harvested and RNA was extracted. Data represent 2 separate experiments. Error bars are standard errors of the mean. (**B**) *pkd2* mRNA expression over 24 h in AX4 developed on KK2 filter paper as measured using RNA-Seq by Parikh A et al. [47]. Image reproduced from data exported from dictyexpress.org [48,49]. To make the RNASeq and qRT-PCR measurements more directly comparable, the log2 of RPKM is plotted.

**Figure 2 cells-13-00610-f002:**
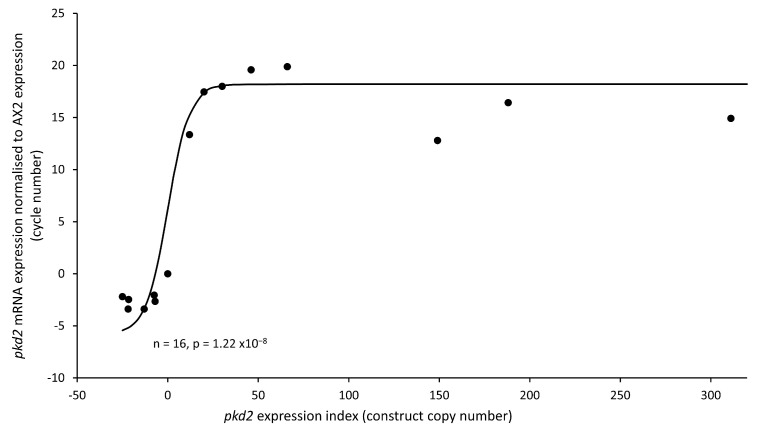
Expression of *pkd2* was altered in the isolated transformants. *pkd2* mRNA expression correlated with plasmid copy number. Individual dots represent the mean data for individual strains for 3 experiments. Expression levels were normalized against the *abpC* mRNA levels to adjust for loading and then measured relative to AX2 expression. The curve was fitted to a sigmoidal (hyperbolic tangent) function by the least-squares method and was highly significant (*p* = 1.22 × 10^−8^, F test). Antisense constructs are represented by negative copy numbers and overexpression constructs are represented by positive copy numbers; this convention has been described previously [38]. AX2 contains no construct so has a copy number of 0.

**Figure 3 cells-13-00610-f003:**
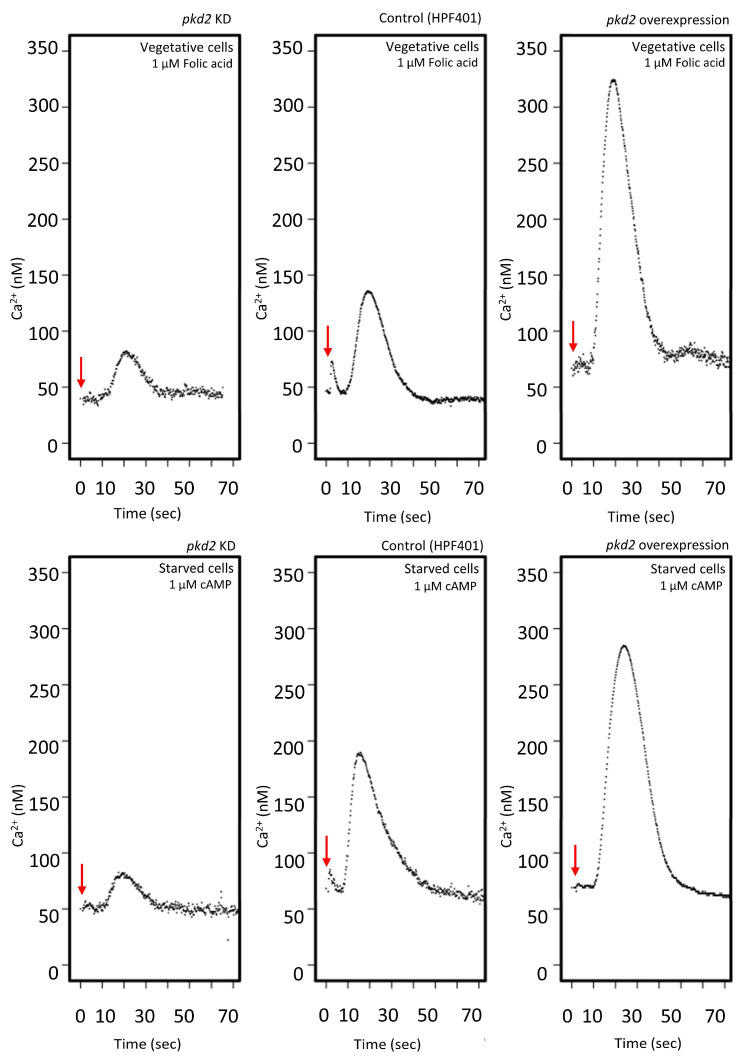
Cytoplasmic Ca^2+^ responses are larger in Polycystin-2*-*overexpressing strains and smaller in knockdown strains. Representative calcium responses in three strains, *pkd2* KD strain HPF644 (copy number 13), control (HPF401) and *pkd2* overexpression strain HPF839 (copy number 311). The top panel illustrates calcium reactions in vegetative cells when stimulated by folic acid, while the bottom panel depicts calcium responses in starved cells (7 h) stimulated with cAMP. Recordings commenced at 0 s, and the stimulus was administered within 1 s (highlighted by red arrows). The minor peak preceding the chemoattractant-induced calcium response results from the mechanical force of the stimulus injection and does not influence the magnitude of the chemoattractant response.

**Figure 4 cells-13-00610-f004:**
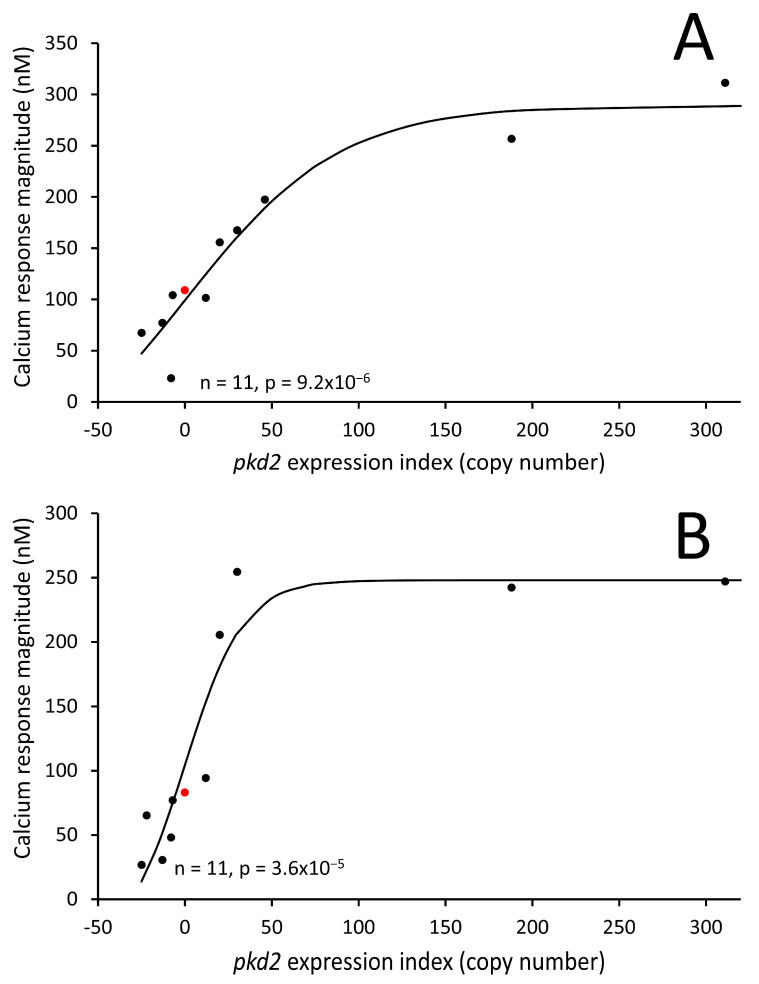
Calcium response magnitudes correlate with plasmid copy number. Cytosolic response magnitudes (Ca^2+^ nM) in strains with different copy numbers for *pkd2* overexpression or KD constructs. Each point is the mean of 2–5 independent recordings for an individual strain. (**A**) Starved cells were stimulated with 1 µM cAMP. (**B**) Vegetative cells were stimulated with 1 µM folic acid, measured from real-time recordings. Response magnitudes positively correlate with the *pkd2* expression index. Control strain HPF401 is indicated in red. Regression lines were fitted to a sigmoidal (hyperbolic tangent) function. The regressions were highly significant with the indicated probabilities (F test).

**Figure 5 cells-13-00610-f005:**
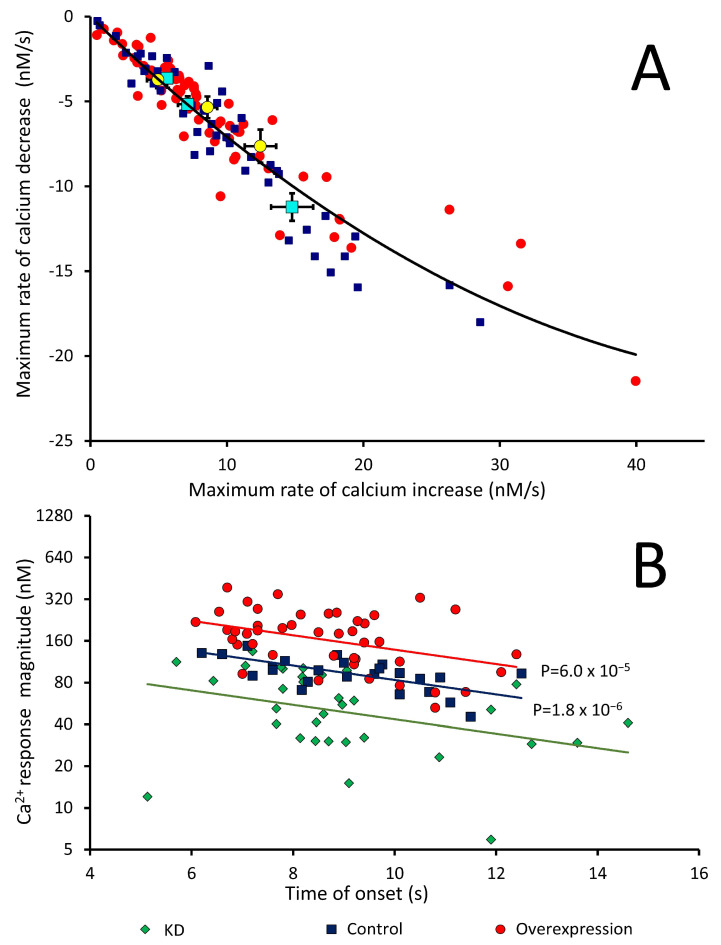
Kinetics of chemotactic Ca^2+^ responses in Polycystin-2 strains. (**A**) Negative relationship between maximum rise and fall rates in control, *pkd2-*overexpressing and knockdown (KD) strains. Each point represents an individual experiment. Blue squares indicate responses to 1 µM cAMP and red circles indicate responses to 1 µM folic acid. Aqua squares are means for cAMP responses and yellow circles are means for folic acid responses from left to right, KD, control and overexpression strains. Rise and fall rates at all stages in the responses were calculated from the Ca^2+^ concentrations at successive time points and the maximum rates of Ca^2+^ increase and decrease were then determined graphically using features of the R statistics and graphics package. Multiple regression analysis showed that neither the attractant (folate or cAMP) nor the strain type (overexpression or KD or control) made a difference in the regression relationship (*p* > 0.05). (**B**) Relationship between the response onset time and magnitude. The relationship is log-linear and negative for *pkd2* overexpression, KD and control strains; however, each set of strains falls on a different line. Multiple regression analysis showed that the intercepts (significance probabilities shown) but not the slopes (*p* > 0.05) differed significantly from the line for the control strain.

**Figure 6 cells-13-00610-f006:**
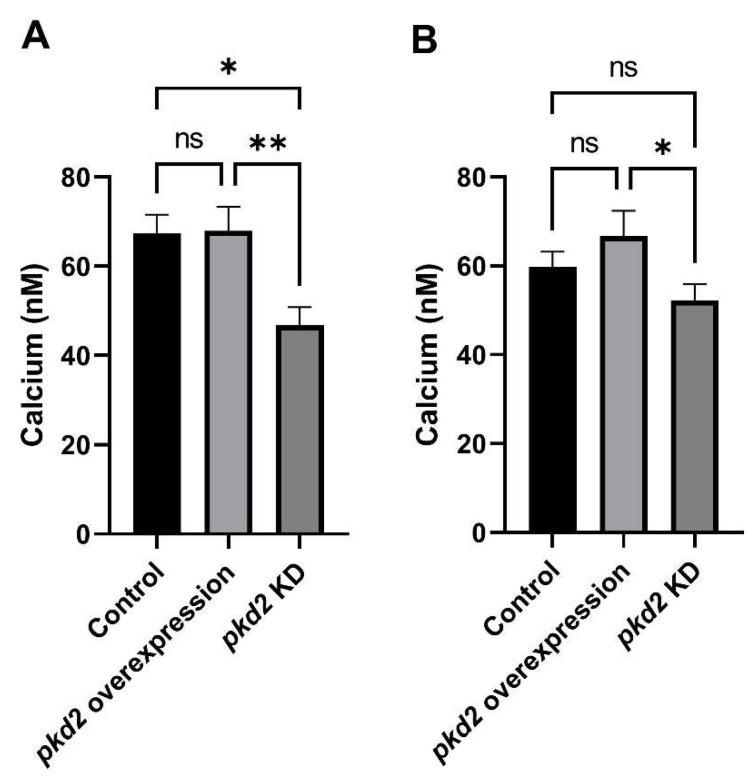
Basal cytosolic calcium concentration is reduced in *pkd2* knockdown transformants. (**A**) Basal [Ca^2+^]_c_ in cells starved and developed for 7 h in MES-DB. There was a significant difference in the basal cytosolic Ca^2+^ levels between the wild-type and KD strains (* *p* = 0.0116, ns = not significant), and the overexpression and KD strains (** *p* = 0.0031, One-Way ANOVA, with pairwise comparisons made by the Least-Squares Difference method). Error bars are standard errors of the mean. KD had six strains (n = 15), overexpression had eight strains (n = 19) and control had HPF401 one strain (n = 12). (**B**) Basal [Ca^2+^]_c_ of vegetative cells growing in HL5 medium. The differences were not significant between wild-type and overexpression transformants, or wild-type and KD transformants. However, the difference between the overexpression and KD transformants was significant (* *p* = 0.0294, ns = not significant, One-Way ANOVA, with pairwise comparisons made by the Least-Squares Difference method). Error bars are standard errors of the mean. KD had six strains (n = 15), overexpression had eight strains (n = 18) and control had HPF401 one strain (n = 13).

**Figure 7 cells-13-00610-f007:**
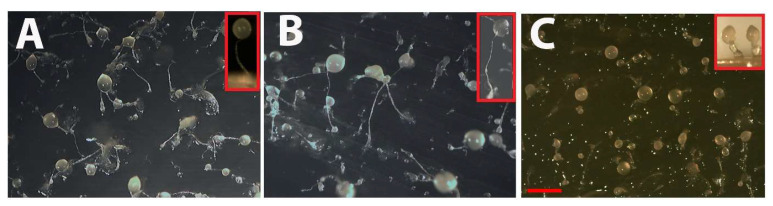
Altering Polycystin-2 expression affects differentiation and multicellular morphogenesis. Effects of Polycystin-2 expression levels on multicellular morphogenesis in *Dictyostelium*. Cells were grown on lawns of *Enterobacter* at 21 °C for ~24 h until mature fruiting bodies had formed. Photographs were captured using an Olympus SZ61 Moticam 2300 camera attached to a dissection microscope from above and from the side (Inserts). (**A**) Wildtype AX2. (**B**) Polycystin-2 KD strain HPF833 (copy number 22) exhibits very thin weak stalks, often unable to support the sorus. Insert (in red boxes) shows side-on image of typical long thin fragile stalks. (**C**) Overexpression strain HPF845 (copy number 149) has short thick stalks, suggesting increased stalk cell differentiation. Insert shows side view of typical fruiting bodies with short, thickened stalks. Scale bar = 1 mm.

**Figure 8 cells-13-00610-f008:**
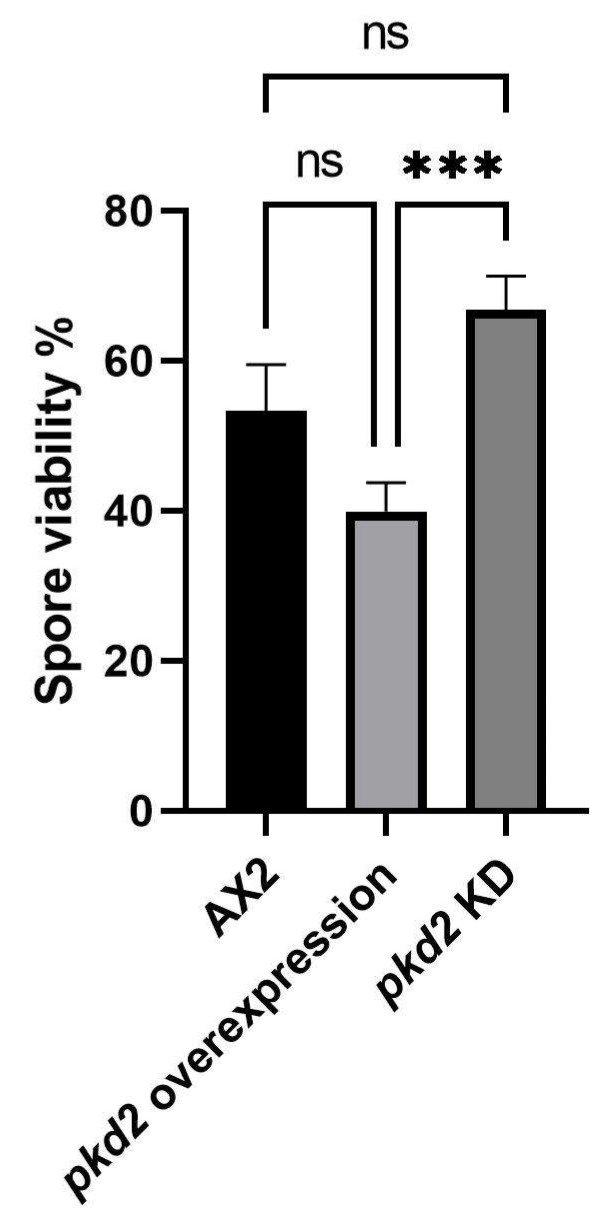
Polycystin-2 affects spore viability. Spores from three Polycystin-2 KD strains and three overexpression strains were harvested from fruiting bodies and heat activated. A known number of spores were then plated on lawns of *Enterobacter* and allowed to germinate. The viability was calculated by determining the number of germinated spores as a percentage of the number of plated spores. The means are calculated from pooled data from all strains within each group over three individual experiments. Error bars are standard errors of the mean. There was a significant difference in the viability between the KD strains and overexpression strains, KD n = 9, overexpression n = 9, control n = 3 (*** *p* = 0.0003, ns = not significant, Unpaired *t*-test).

**Figure 9 cells-13-00610-f009:**
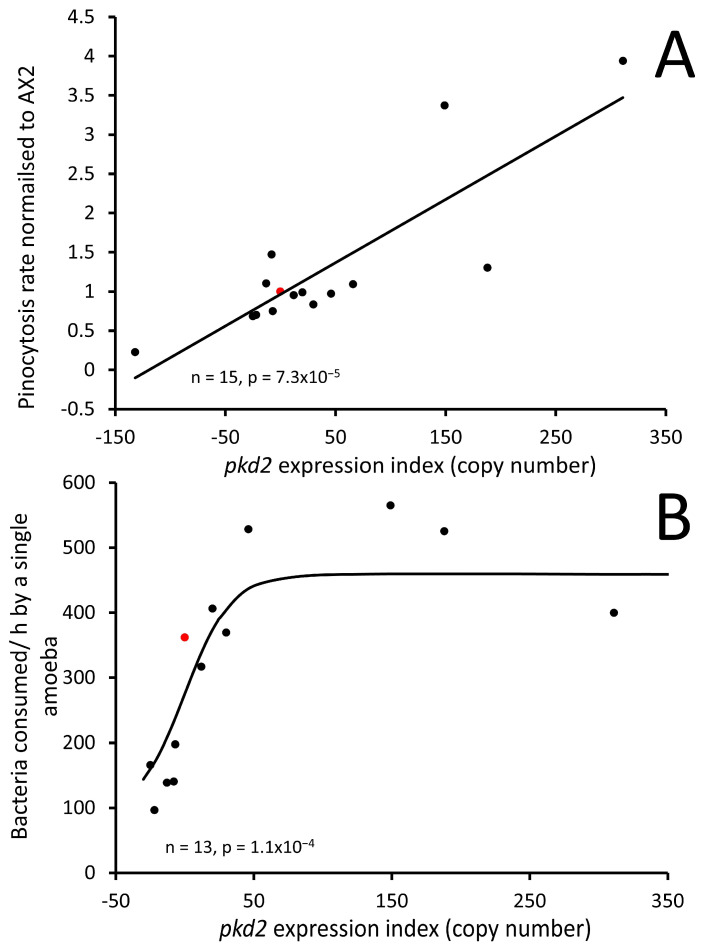
Polycystin-2 positively regulates pinocytosis and phagocytosis. (**A**) Pinocytosis was positively regulated by Polycystin-2. The consumption rate of FITC dextran-HL5 medium was measured in AX2 (red marker), Polycystin-2 KD and overexpression strains. The regression line was fitted by least squares to a linear function, and was significant, probability indicated (F test). (**B**) Phagocytosis was measured as consumption of *E. coli* expressing the red fluorescent protein DsRed. Phagocytosis and pinocytosis rates increased as Polycystin-2 expression increased, AX2 indicated in red. The regression line was fitted to a sigmoidal (hyperbolic tangent) function and was highly significant, probability indicated (F test).

**Figure 10 cells-13-00610-f010:**
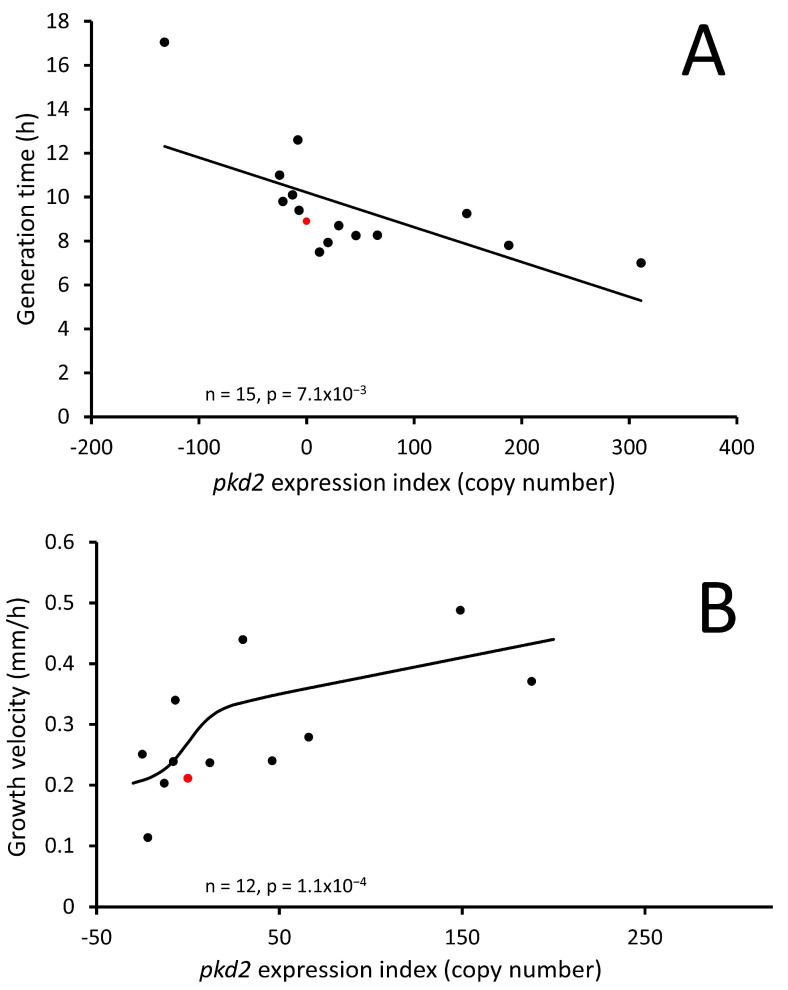
Polycystin-2 positively regulates *Dictyostelium* growth rates in HL5 medium and on lawns of *E. coli*. (**A**) Expression of Polycystin-2 is significantly correlated with generation times in liquid medium, longer generation times indicate slower growth. AX2 is indicated in red. Strains were grown exponentially in HL5 medium and shaken at 150 rpm, and generation times were determined by log-linear regression of the relationship between cell density and time. The regression line was fitted by least squares to a linear function. (**B**) Expression of Polycystin-2 is significantly correlated with growth rates on bacterial lawns. The growth velocity (rate of plaque expansion) was measured from linear regressions of the diameter of the plaques vs. time (h). AX2 is indicated in red. The regression line was fitted to a modified sigmoidal (modified hyperbolic tangent) function. The regressions were highly significant with the indicated probabilities (F test).

**Figure 11 cells-13-00610-f011:**
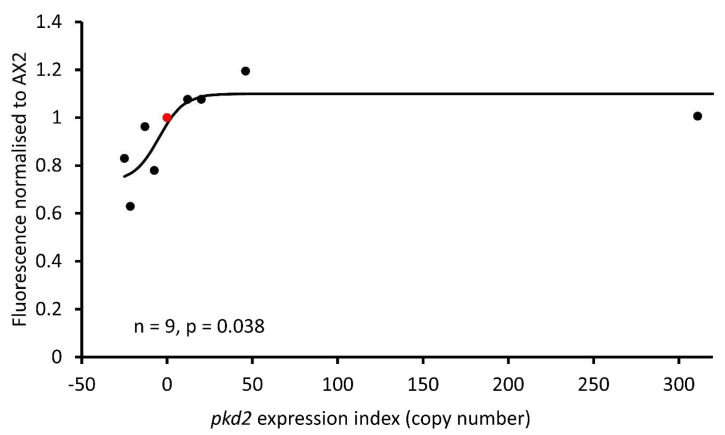
Polycystin-2 expression affects the mass or pH of acidic vesicles. LysoSensor^TM^ Blue DND-167 was used to stain vegetative cells, and the increase in fluorescence was measured in a CLARIOstar plate reader. Fluorescence increased as Polycystin-2 expression increased, AX2 indicated in red. The regression line was fitted to a sigmoidal (hyperbolic tangent) function and was significant with the indicated probability (F test). Four overexpression, four KD strains and AX2 were tested in four independent experiments, and the data were normalized to AX2 within each experiment.

## Data Availability

The data presented in this study are available upon request from the corresponding author.

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
