# Peer review of "Polycystin-2 Mediated Calcium Signalling in the Dictyostelium Model for Autosomal Dominant Polycystic Kidney Disease"

_cells, 2024, doi:10.3390/cells13070610_

Round 1

Reviewer 1 Report

Comments and Suggestions for Authors

The manuscript “Polycystin-2 mediated calcium signalling in the Dictyostelium 2

model for Autosomal Dominant Polycystic Kidney Disease” studied cellular-level function of Polycystin2 (PC2) using Dictyostelium.  Mutation in Polycystin2 is a major genetic factor underlying polycystic kindney disease, however its cell biological role is not at all well understood including how it relates to abnormal cyst formation.  The authors have taken an interesting approach using Dictyostelium to vary the degree of overexpression and knockdown  in clones harboring different number of plasmid constructs.  The main finding is that basal cytosolic Ca2+ as well as transient upon chemoattractant stimuli are either enhanced or suppressed according to the level of PC2 expression.  The work also clarified growth and developmental defects that were previously overlooked.  The introduction is extremely well written with plenty of background information.  The results are new and important and I recommend publication after addressing a few points listed below.

According to Fig. 1, there is developmental change in the expression level of PC2 between 8 and 12 hrs.  It’s counterintuitive that authors chose to study only 0hr and 7hr cells where expression of PC2 appears to be minimal.  While I do understand that the cells are differentiated differently for RT-PCR and Ca2+ experiments, why not 4 and 12 hrs when may be maximally expressed? 

Fig. 1B is is an image reproduction and does not really reflect the original data.  Can authors retrieve this from the database.  It is very awkward that this has to be presented in this manner and goes against legitimate data management.

About the Ca2+ response, is Fig 3 showing representative data from vegetative cells?  I may have missed something here.  The starved cells data are described in main text but I don’t think the data are shown. Will it be possible to provide an overlay in a single panel with using different marks (+, - * etc)?    Are these two data merged in fig 4?  If so, please specify.  There may be a discrepancy between the present results and those reported earlier. 

About Fig. 7 and 8.  The developmental phenotype is quite intriguing.  In Line427, the authors claim that resting level of Ca2+ is already maximal in the wildtype strain, yet the authors observe stalky phenotype in the overexpressor.   Does this imply there are specific signal induced Ca2+ transients involved in cell differentiation?  Is there support for this in earlier literatures?  Please elaborate.

Minor points:

Fig. 10 lacks the data point for 300 copy number.

L512  DS-red -> DsRed

Reviewer 2 Report

Comments and Suggestions for Authors

In this study, the authors investigated polycystin-2 mediated calcium signaling in the model organism Dictyostelium discoideum. The results showed that chemoattractant-stimulated cytosolic calcium responses were increased in overexpression strains and decreased in knockdown strains. The kinetics analysis suggested that Polycystin-2 significantly contributes to the control of calcium responses. Additionally, basal cytosolic calcium levels were found to be reduced in Polycystin-2 knockdown transformants. These changes in calcium signaling also affected various downstream processes, such as growth rates, endocytosis, stalk cell differentiation, and spore viability.

In Dictyostelium, previous research has already been conducted to elucidate the mechanisms regulating Polycystin-2, as described by the authors. The additional data presented in this study serve to complement and enhance our existing understanding of these regulatory mechanism.

Major revision

1- Typically, development is carried out in agar P, and variations in agar concentration (1-1,5%) and composition (in this case, in water) play a crucial role in determining the timing of different developmental phases. Therefore, I would appreciate clarification from the authors on why they chose water agar and whether they should provide a time course to compare timing and development. Additionally, in Figure 1, gene expression trends are compared. However, due to the different measurement types on the y-axis, the statement in line 270 is unclear. In panel A, during the first 8 hours (excluding the peak at 2 hours), gene expression is reported to be downregulated. The authors should clarify this aspect, especially regarding the statement in lines 272-274, which appears to be in contrast with the data previously published in reference 31.

2- Considering that the results integrate our existing knowledge about the role of PDK2 in calcium regulation, an innovative aspect of the study that would distinguish it from previous work could be to determine the putative partners of  polycystin-2 (as suggest in lines 638-640). I would recommend conducting this experiment to provide clarity. 

Minor revision

1- The caption for Figure 2 needs to be shortened. Lines 304-307 appear to describe a section more in-depth, better suited for the Results section

2-In Figure 3, I would suggest indicating the point of pulse addition with an arrows

3- Line 578-580 Referring to the protein is an inference, as outlined by the results where clones are isolated based on gene expression levels. In my opinion, the authors should consider rephrasing of the sentence.

Round 2

Reviewer 2 Report

Comments and Suggestions for Authors

The authors have provided thorough responses to the queries raised during the initial review and have adequately justified their inability to conduct further experiments. Therefore, I believe the manuscript is suitable for publication in the Cells journal.

Author Response

These are the reviewer comments, therefore we require no further action for this reviewer:

Comments and Suggestions for Authors

The authors have provided thorough responses to the queries raised during the initial review and have adequately justified their inability to conduct further experiments. Therefore, I believe the manuscript is suitable for publication in the Cells journal.